# Geo-Referenced Mapping through an Anti-Collision Radar Aboard an Unmanned Aerial System

**Lapo Miccinesi** ID**, Luca Bigazzi** ID**, Tommaso Consumi, Massimiliano Pieraccini** * ID**, Alessandra Beni** ID**, Enrico Boni** ID **and Michele Basso** ID

Department of Information Engineering, University of Florence, Via Santa Marta 3, 50139 Firenze, Italy; lapo.miccinesi@unifi.it (L.M.); luca.bigazzi@unifi.it (L.B.); tommaso.consumi@unifi.it (T.C.); alessandra.beni@unifi.it (A.B.); enrico.boni@unifi.it (E.B.); michele.basso@unifi.it (M.B.)
* Correspondence: massimiliano.pieraccini@unifi.it

**Abstract:** Unmanned aerial systems (UASs) have enormous potential in many fields of application, especially when used in combination with autonomous guidance. An open challenge for safe autonomous flight is to rely on a mapping system for local positioning and obstacle avoidance. In this article, the authors propose a radar-based mapping system both for obstacle detection and for path planning. The radar equipment used is a single-chip device originally developed for automotive applications that has good resolution in azimuth, but poor resolution in elevation. This limitation can be critical for UAS application, and it must be considered for obstacle-avoidance maneuvers and for autonomous path-planning selection. However, the radar-mapping system proposed in this paper was successfully tested in the following different scenarios: a single metallic target in grass, a vegetated scenario, and in the close proximity of a ruined building.

**Keywords:** anti-collision radar; autonomous flight unmanned aerial system; obstacle detection; obstacle avoidance; radar mapping; unmanned aerial system (UAS); unmanned aerial vehicle (UAV)

## 1. Introduction

Recent technological advances made in unmanned aerial systems (UASs), especially in combination with autonomous flights, could have a major impact on civilian tasks [1]. For instance, they could not only enable the provision of social services, such as the delivery of goods, drugs and sanitary equipment, but could also perform environmental monitoring in remote regions [2]. Since these applications would involve flights performed in complex outdoor scenarios, capabilities for autonomous obstacle detection and avoidance are of paramount importance. Researchers have recently devoted great effort to the challenging task of autonomous UAS flights [3].

An open challenge for autonomous flights is to develop a reliable mapping system of the surrounding environment. A mapping procedure of the surrounding area would not only enable the detection and avoidance of obstacles, but also the elaboration of effective decommitment strategies.

Currently, vision systems are often used with this aim [4–6]. However, their maximum detectable distance (usually lower than tens of meters) is a serious drawback [7], which does not make them the optimal solution for mapping outdoor environments. Indeed, for a UAS flying at a relatively high speed, this distance may not be sufficient for implementing a suitable decommitment strategy. Moreover, optical sensors are dramatically affected by light exposure and weather conditions.

Another way to detect possible obstacles and overcome limitations due to weather conditions is to use radar equipment. In fact, radars allow for the extension of the maximum distance of detection and are only slightly affected by environmental conditions (e.g., light, fog and rain). Given the advantages provided by radar sensors, many research groups have already worked in this direction.

In [8] Sacco G. et al. proposed a MISO (multiple-input single-output) system based on an FMCW radar, which worked at 24 GHz and was specifically optimized for drone detection. The transmitting and receiving serial arrays of patch antennas have been suitably designed to operate up to 150 m. Experimental tests performed on the ground in a controlled scenario evidenced the correct estimation of the target position in the range–azimuth plane.

Recently, a new class of radars has been developed for obstacle avoidance in automotive applications. These radars are single-chip devices that work at high frequency (W-band [9]). Today, these sensors are largely employed in the automotive field, but also for indoor mapping through terrestrial rovers.

S. Dogru et al. [10] studied the mapping performance of a radar in indoor environments to support mobile robots employed in search and rescue operations in low visibility areas. They used an FMCW radar working in the frequency range 76–81 GHz, with two transmitting and three receiving antennas to construct two-dimensional maps of the surrounding area. The radar was mounted on a robot, which moved across the investigated area. The quality of radar-based maps was compared to that of light detection and ranging (LIDAR) maps by using a quantitative map-quality metric. Their results evidence that even though LIDAR still outperforms radar, when mapping low visibility environments, i.e., with fog or smoke, radar provides better results in terms of mapping.

In [11], S. Lee et al. illustrated the mapping results of an indoor environment obtained with a dual-mode radar. The sensor was a multiple-input multiple-output (MIMO) system working at a central frequency of 62 GHz. It was capable of alternatively transmitting two waveforms with different bandwidths, optimized for long-range and short-range detection, respectively. The radar was mounted on a robot that moved in the area to be mapped and sent information about its own position. By combining the radar data with the information for the robot's position, they successfully constructed a map of the surrounding environment.

The small size and light weight of these millimeter-wave radars make them promising candidates for operating aboard a drone. Some research groups have already investigated this possibility [12–14]. These works also present strategies to overcome and mitigate the problem of poor elevation resolution, a common characteristic of these radars.

Authors of [12] implemented an active drone detection system. They mounted a millimeter-wave radar on a drone with the aim of detecting, tracking, and pursuing other target drones. Although the tracking radar performed only 2D measurements, the 3D data are recovered by complex maneuvering the pursuer drone. Despite the promising results, the proposed technique has limited applications as it assumes the target drone as the only other airborne object and requires the drone to perform specific complex maneuvers.

A millimeter-wave MIMO radar capable of three-dimensional sensing for applications in UAS formation flights and obstacle avoidance is presented in [13]. Specifically, commercially derived millimeter-wave radar technology was integrated with a custom MIMO antenna array that was optimized for specific flight dynamics. The results of the experimental test were promising, as the system confirmed the three-dimensional detection capabilities of the investigated target.

Authors of [14] present a system for mapping the environment surrounding a UAS flight that is based on the fusion of a millimeter-wave radar and a monocular camera. Specifically, monocular camera data are used to provide a reference for mapping, and to identify targets in the surrounding environment. Then, by using an extended Kalman filter, the radar data are fused with that of the camera, thus enabling the local mapping of targets. A possible drawback of this system is that the reference used for mapping is provided by the monocular camera itself. Therefore, in certain scenarios, the system may lose this reference, thus, leading to possible errors, which may compromise the mapping process.

Therefore, the aim of this article is to present some preliminary tests of a radar aboard a UAS that is able to both operate as obstacle-avoidance equipment, and is able to provide reliable mapping of its surrounding area that could be used to elaborate effective de-

commitment strategies. This is a challenging goal that other radar systems presented in scientific literature are not designed to accomplish.

The single-chip radar, mounted aboard the drone, provides the direction of arrival (DOA) of the scattered signal [15], and the detected targets can be correctly disposed on a local map by using the pose (position and attitude) provided by the UAS flight controller. The system was developed in the robot operating system (ROS) environment, so as to enable integration with other sensors [16]. The complete system was experimentally tested in a controlled scenario, with a single target, and in the following complex scenarios: in a wood and in close proximity to a ruined building. This article also proposes a path-planning strategy for taking into account the poor elevation resolution of this kind of radar.

## 2. Materials and Methods

The mapping system was developed in the ROS environment in order to be easily integrated with other sensors or to be implemented on different UAS platforms. A block scheme of ROS architecture is shown in Figure 1. The key task is the *3D mapping* node, which converts the position of detected targets from the radar frame to the fixed frame and processes the voxel map. This node communicates with the physical sensors through the *Radar* node and *Telemetry* node. The *Radar* node handles the radar and publishes the information about the detected targets on a proper topic (for example position, speed, signal amplitude). The drone interfaces with the *3D mapping* node through the *Telemetry* node. This node publishes real time kinematic (RTK) position and attitude (quaternion) in standard ROS-telemetry messages. It is noteworthy that the whole system is designed to work in real time, so it is able to provide timely alerts or the re-planning of paths.

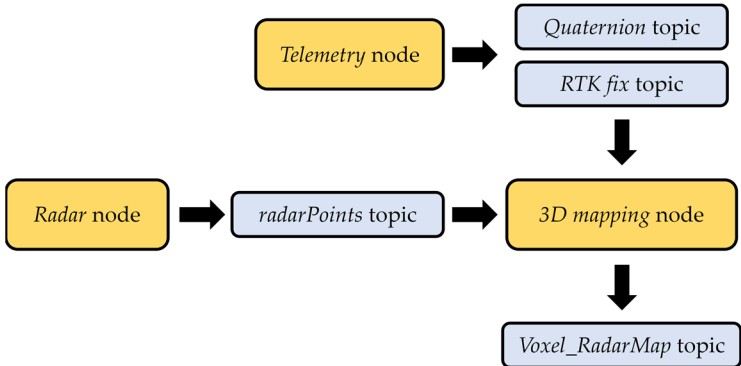

**Figure 1.** Block scheme of ROS architecture.

The functional scheme is shown in Figure 2. This scheme is independent from the specific drone, as it allows for the system to be easily changed by maintaining the same ROS messages. Therefore, the mapping system could be implemented on different UASs with minor changes.

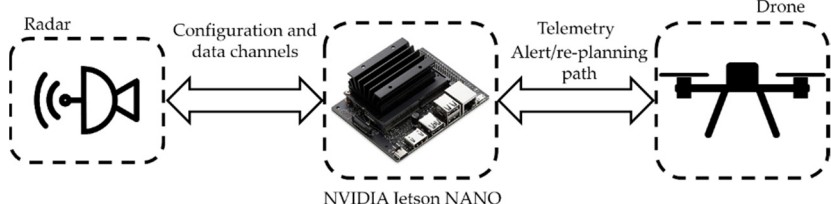

**Figure 2.** Functional scheme of the proposed architecture.

The ROS workspace was implemented on a NVIDIA Jetson Nano board. This board was connected to the drone to obtain the telemetry data through the *Telemetry* node, and to provide alerts or path re-planning to the drone pilot (not yet implemented). The radar was connected to the computer board, and it was controlled by the *Radar* node. The *Radar* node

was used for configuring and receiving data from the radar. The NVIDIA Jetson Nano was also used for the *3D mapping* node. Indeed, the 3D mapping algorithm could be very expensive in computational terms.

### 2.1. 3D Mapping Node

The 3D map was created in the *3D mapping* ROS node. Within this node, both the Telemetry messages, coming from the drone, and the position of detected objects, from the radar, were received. Each object's position was moved from the radar frame to the fixed frame. Subsequently, it was mapped in a voxel map using the OctoMap library [5,17]. Indeed, the position of the detected targets was referred to the radar frame as shown in Figure 3.

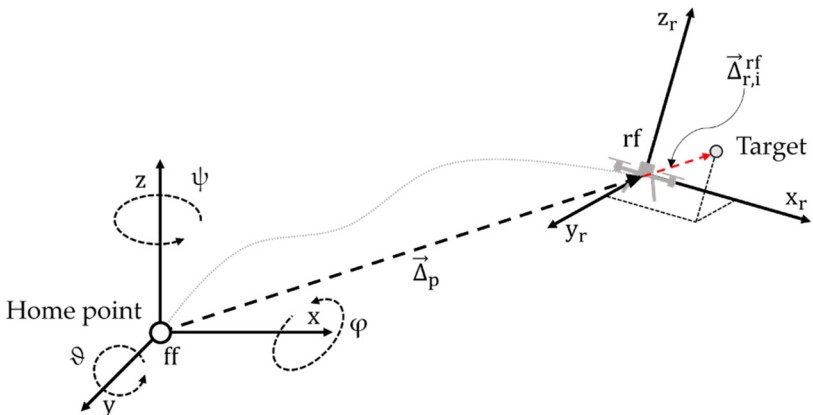

**Figure 3.** The image shows the two reference frames, where ff is the fixed frame and rf represents the radar frame.

The fixed reference frame has its origin corresponding to the take-off point (or home point). The orientation of the fixed frame is also coherent with the attitude quaternion provided by the *Telemetry* node during the initialization phase.

The radar reference frame is fixed with respect to the drone as shown in Figure 3. This means that the radar frame origin and orientation depend on the traveled trajectory of the drone during the mission.

First, to correctly map the radar target in the fixed frame, it is necessary to obtain the local path travelled by the drone during the mission in cartesian coordinates. Indeed, the Global Navigation Satellite System (GNSS) provides the position in terms of latitude and longitude. Equation (1) defines the difference between the latitudinal and longitudinal coordinates of the take-off point and the current ones.

$$\begin{cases} \Delta_\alpha = \alpha_c - \alpha_h \\ \Delta_\beta = \beta_c - \beta_h \end{cases} \tag{1}$$

where $\alpha_c$ and $\alpha_h$ are the latitude of the current position and the home point, while $\beta_c$ and $\beta_h$ represent the longitude of the current position and the home point, respectively.

To convert the trajectory from latitude-longitude coordinates into cartesian coordinates, it is possible to use the relations (2).

$$\vec{\Delta}_P = \begin{cases} \Delta_x = r \times \sin \Delta_\beta \times \cos \alpha_c \\ \Delta_y = r \times \sin \Delta_\alpha \\ \Delta_z = z_c - z_h \end{cases} \tag{2}$$

where r is the average radius of the Earth, and $(\Delta_x, \Delta_y, \Delta_z)$ are the coordinates in the fixed frame of Figure 3. The relative height of the drone is given only by the difference between the initial and the current height, which are both provided by the GNSS. In fact, the GNSS system provides height in respect to sea level in meters.

The radar frame and fixed frame also have different orientations. For this reason, we can consider the complete rotation matrix, $R_{xyz}(\vartheta, \varphi, \psi)^{-1}$, to orient the detected targets as a function of the fixed frame:

$$R_{xyz}(\vartheta, \varphi, \psi)^{-1} = R_{xyz}(\vartheta, \varphi, \psi)^{T} = \begin{bmatrix} c_\vartheta c_\psi & -c_\vartheta s_\psi & s_\vartheta \\ c_\varphi s_\psi + s_\vartheta c_\psi s_\varphi & c_\varphi c_\psi - s_\varphi s_\vartheta s_\psi & -s_\varphi c_\vartheta \\ s_\varphi s_\psi - c_\varphi s_\vartheta c_\psi & s_\varphi c_\psi + c_\varphi s_\vartheta s_\psi & c_\varphi c_\vartheta \end{bmatrix}^{T} \tag{3}$$

where $\vartheta$, $\varphi$ and $\psi$ are the pitch angle, the roll, and the heading angle, respectively, and the symbols c and s are the cosine and sine of the respective angle. This rotation matrix was evaluated by considering the axis of the selected drone according to the right-hand rule.

General speaking, the radar is not able to provide both the azimuth and the elevation of the target. In this specific case we decided to use the azimuth resolution and to always consider zero as the elevation of the object ($z_{r,i} = 0$ m). This is equivalent to assuming that each target is on a horizontal plane at the same height of the drone. This hypothesis is not as strong as it seems, because usually the target at $z_{r,i} = 0$ m is the most reflective.

Under this hypothesis the rotation in (3) can be reduced to a matrix that considers only the rotation along the heading angle:

$$R_z(\psi)^{-1} = R_z(\psi)^{T} = \begin{bmatrix} c_\psi & -s_\psi & 0 \\ s_\psi & c_\psi & 0 \\ 0 & 0 & 1 \end{bmatrix}^{T} \tag{4}$$

where $c_\psi$ and $s_\psi$ are the cosine and sine of the yaw angle.

Finally, to obtain the coordinates of the targets referred to the fixed frame, it is necessary to consider the rotation (4) and the offset $\vec{\Delta}_p$. Equation (5) shows the complete equation of the desired target position in fixed frame, $\vec{\Delta}_{r,i}^{ff}$:

$$\vec{\Delta}_{r,i}^{ff} = R_z(\psi)^{-1} \times \vec{\Delta}_{r,i}^{rf} + \vec{\Delta}_p \tag{5}$$

where $\vec{\Delta}_{r,i}^{rf}$ is the object coordinates in radar frame:

$$\vec{\Delta}_{r,i}^{rf} = \begin{bmatrix} x_{r,i} \\ y_{r,i} \\ z_{r,i} \end{bmatrix} \tag{6}$$

The target coordinates obtained in (5) could be located on the same map using OctoMap library. This library allows the creation of a voxel map, where each voxel has the coordinates of the detected target defined in the fixed frame. Since the OctoMap library provides functions to search for occupied points, it is possible to use the radar map as an anti-collision and obstacle-avoidance system.

### 2.2. The Radar Sensor

The radar used for this article is an AWR1843BOOST by Texas Instruments [18] (Figure 4). A radar detects the distance of the target by sending and receiving an electromagnetic signal through at least a couple of antennas. Using a MIMO array, it is also able to retrieve the direction of arrival.

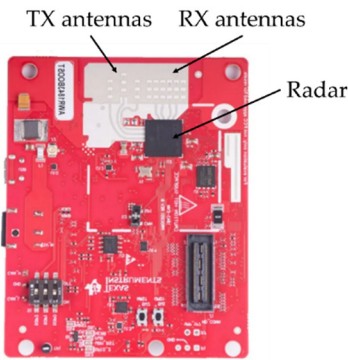

**Figure 4.** AWR1843BOOST by Texas Instruments, US.

The sensor used is comprised of 3 TX and 4 RX antennas, which correspond to 12 virtual antennas, disposed as shown in Figure 5. In Figure 5, the z axis represents the altitude, while the x axis is left to right, and λ is the wavelength of the electromagnetic signal. This arrangement of virtual antennas achieves a good azimuth resolution and a poor elevation resolution [19]. Indeed, resolution is related to the inverse of the z-distance between the antennas.

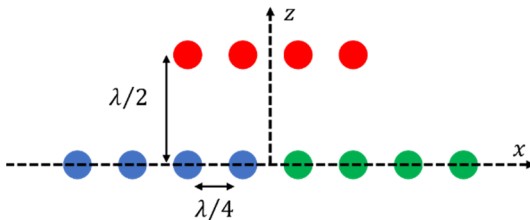

**Figure 5.** Virtual antennas position of AWR1843BOOST [18].

For the current application, the elevation resolution is used only as angular cut-off. In other words, the value of elevation measured by the radar is not used for mapping and it was fixed equal to zero for each target, but it is used as a spatial filter for rejecting the target outside a selected angular area. We set the angular field as ±45 deg in azimuth and [0, 20] deg in elevation. Therefore, all targets outside this interval are not used for mapping.

The radar provides a frequency-modulated signal from 77 GHz to 81 GHz (the whole bandwidth was not used for the experiments) from each TX antenna and registers the echo from all RX as Figure 6 shows. The frequency sweeps (chirps) are collected in a structure called frame.

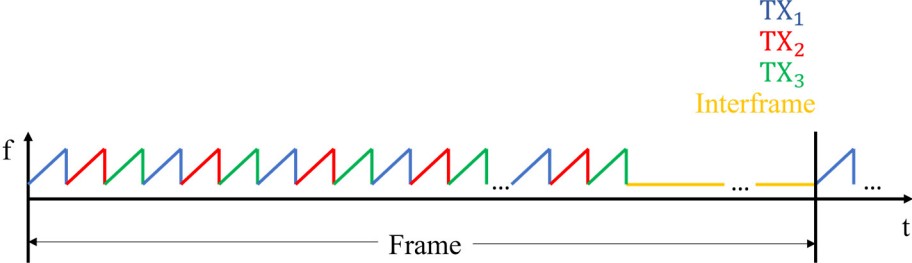

**Figure 6.** Example of a signal provided by AWR1843BOOST [18].

The radar is able to retrieve the position of detected targets using an internal computational unit. Indeed, all the operations in Figure 7 were carried out by the hardware and the library onboard the radar. For each single chirp, the range of the fast Fourier transform (FFT) is calculated for retrieving the distance of possible targets. All the range

FFTs are collected in a matrix. Another FFT was calculated along the doppler direction of the range-FFT matrix. This FFT, called doppler FFT, is used for detecting the target speed. Here, the targets are confused with clutter and noise. To discriminate the targets from false alarms, a two constant false alarm rate (CFAR) processes is applied. The CFAR is an algorithm that, using a threshold, compares the amplitude of each single pixel with the average amplitude of the neighbors [20,21] and selects the pixels over the threshold. For range direction, we used a cell-average smallest of (CASO)-CFAR algorithm [21]. Indeed, the CASO-CFAR is particularly suitable for detecting objects surrounded by clutter. For the doppler direction, a cell-average (CA)-CFAR algorithm [20] was used. The CA-CFAR in doppler direction allows for the selection of targets with well-defined speed. Finally, the position of targets is estimated by considering the antenna pattern. This estimation is performed on the target that exceeds the threshold of both CFAR algorithms to reduce computational load.

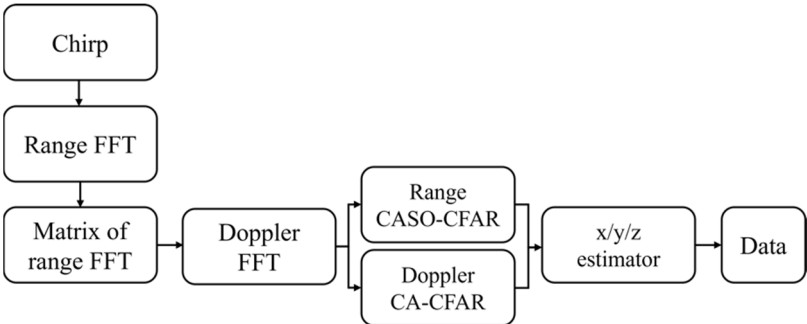

**Figure 7.** Detection chain.

For each detected target, the radar provides x/y/z position, speed, signal-to-noise ratio of CFAR and target index.

The radar was connected to the NVIDIA Jetson Nano through a USB cable with the following two separate communication channels: a configuration port and a data port. The *Radar* node publishes a topic for each detected point with all the information reported above and the time stamp of the frame. It is important to note that the topic is published only if at least one target is detected. The radar topic is subscribed by the mapping node, as described in Section 2.1.

### 2.3. The UAS and Telemetry Node

As explained in Section 2.1, the radar map is produced by considering the GNSS position of the drone. For a reliable map, the authors used a real-time kinematic positioning (RTK) system. Indeed, this system is able to provide the position with an accuracy of about 50 mm, which is enough for radar mapping (the typical range resolution is about 500 mm).

The GNSS-RTK was equipped on a DJI Matrice300RTK and provides the position at 5 Hz. The drone also provides the filtered quaternion, at 100 Hz, for retrieving the attitude.

This drone can communicate with an external computer using an UART port and the C++/ROS library provided by DJI. Using this library, it is possible to access navigation data (telemetry, battery, and navigation status, etc.) and to implement some basic piloting commands, e.g., it is possible to set new waypoints.

The NVIDIA Jetson Nano, with the DJI library, was installed on the drone as shown in Figure 8. The radar was located below the drone, and it was locked to maintain the same orientation as the drone.

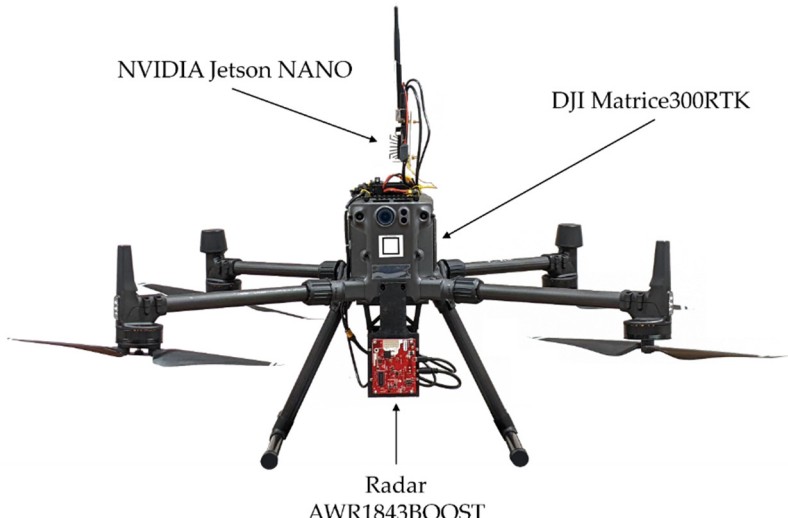

**Figure 8.** DJI Matrice300RTK provided by the radar AWR1843BOOST and by NVIDIA Jetson Nano.

The *Telemetry* node is a modified version of the one provided by DJI. Indeed, we have generalized the DJI telemetry messages by converting them into standard telemetry ROS messages.

### 2.4. Path-Planning Strategy

Using this mapping system, it is also possible to design an avoidance strategy based on radar data. Furthermore, as the map is georeferenced, it could be used also for planning the return path by considering the obstacles detected in the outbound flight. Figure 9 shows the proposed obstacle-avoidance strategy, based on radar map. There are two possible maneuvers, as follows: (1) go around sideways, (2) fly over (see Figure 9). Fly below is a forbidden maneuver. A word of caution needs to be given regarding the second maneuver (fly over). The radar has poor elevation resolution, so for prudential reasons we assume that any detected target is positioned at the same altitude of the drone. When the drone increases the altitude, it continues to detect the obstacle as long as it is inside the vertical view-angle of the radar. Therefore, a single target could be represented by a sort of vertical column, but this is not a problem in path planning.

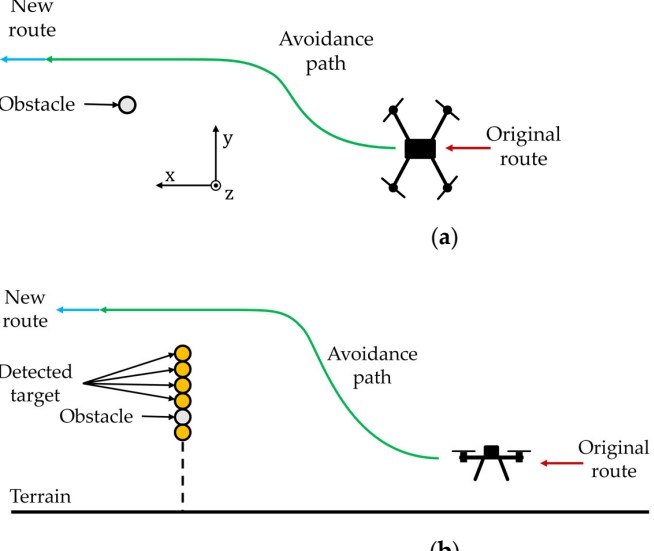

**Figure 9.** Proposed avoidance strategies: (**a**) go around sideways, (**b**) fly over.

## 3. Results

The equipment was tested in a controlled scenario with only one electromagnetic target (a naval corner reflector) and in the following two realistic scenarios: a small wooded area and an almost ruined building.

The radar parameters were the same for all the tests. The maximum range was fixed at 120 m, with 0.5 m of range resolution. The azimuth angle was within $\pm 45°$, and the elevation within $[0, 20]$ deg. This value was selected to filter out possible artifacts due to radar side lobes or other sources of noise. The maximum speed that the radar was able to detect was $\pm 10.4$ m/s, with speed resolution of 0.31 m/s. The frame periodicity was 10 Hz, but the radar shares its topic only if at least one target is detected.

### 3.1. Controlled Scenario

Figure 10 shows the setup used during the test in a controlled scenario. The target was at 68 m in front of the drone. It is important to note that the target was located on a small hill and difference of the altitude was about 3 m. The drone path is shown in Figure 11. The drone flew towards the target and made a lateral movement, first going towards the left and after towards the right. After this lateral movement, the drone flew back to the home point. As shown in Figure 11, altitude was also changed during the flight.

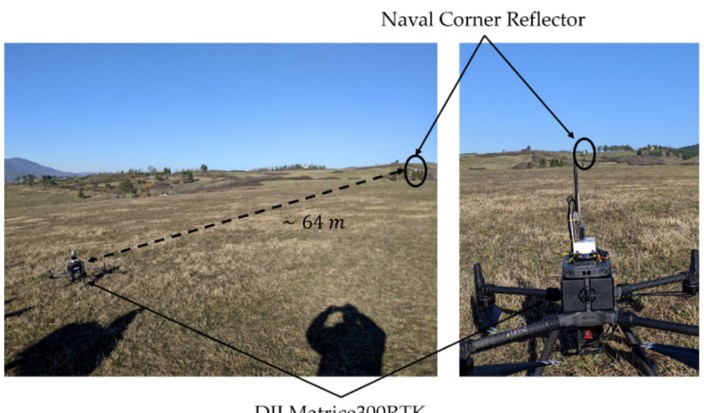

**Figure 10.** Picture of the controlled scenario.

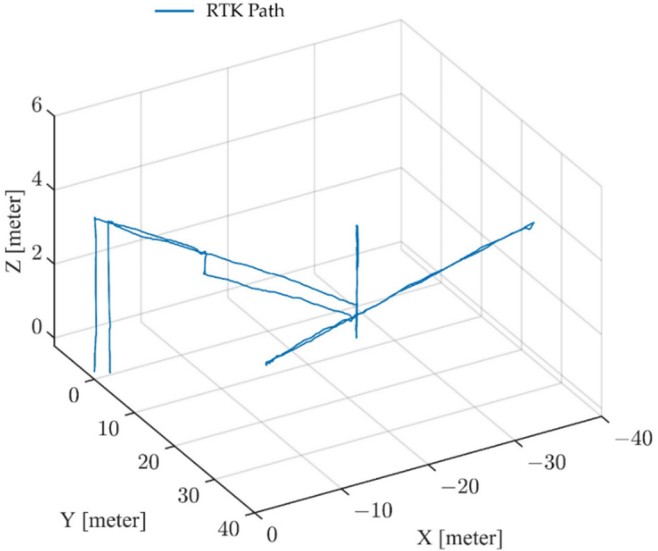

**Figure 11.** Three-dimensional path of drone in fixed frame.

Figure 12 shows the detected target in the radar frame during the whole flight. We can notice a series of targets that come close to the drone starting form $Y_{radar} = 70$ m to 30 m (spotlight with orange). The group of targets at 30 m corresponds to the lateral movement. These signals in orange area correspond to the naval corner reflector. Hence, it is evident that a single target in a radar frame could be seen as a "wall" in front of the drone.

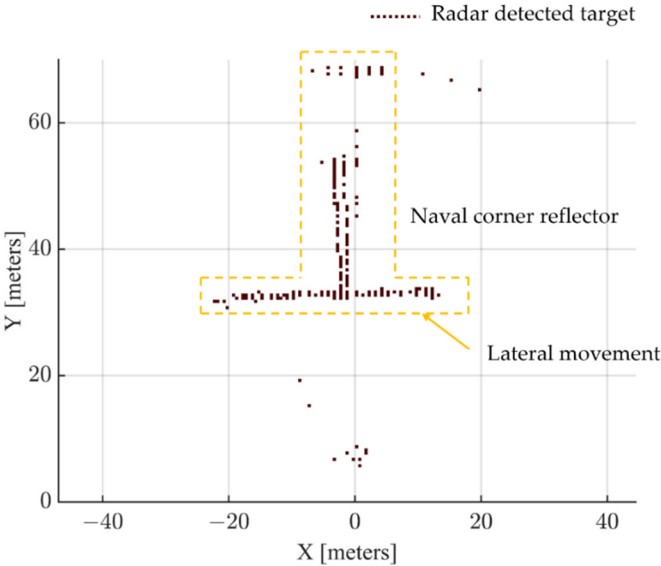

**Figure 12.** Detected target in radar frame.

The map in Figure 13 was evaluated using (6) and rotation matrix (5). Most of the targets in the fixed frame as shown in Figure 12 are grouped in a cluster that corresponds to the corner reflector. The other sparse targets visible on the map are probably related to the ground.

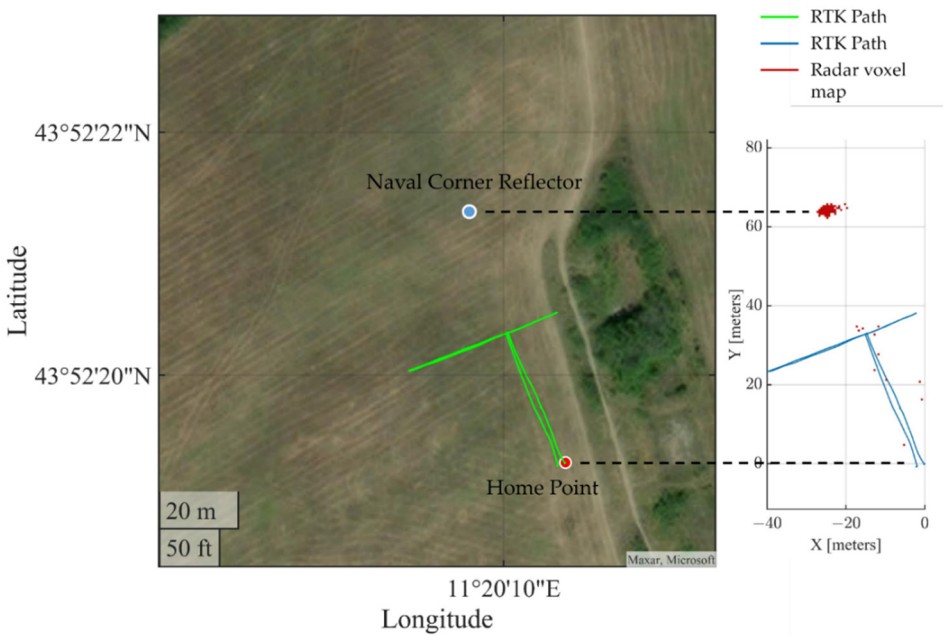

**Figure 13.** Radar map in fixed frame seen from above.

The vertical profile of the map is shown in Figure 14. As described in Section 2.3, the objects were mapped using only the drone's altitude. It is interesting to note the sort of spurious targets along the path. These are a consequence of the fact that the detected

targets are always positioned at the flight altitude, and also when the radar detects ground targets at the lowest edge of the view cone. This is a prudential measure, but it has the unavoidable drawback of producing these false obstacles along the path. This is not a critical problem for the following two reasons: (1) these spurious targets appear only at very low altitudes, where the drone should not normally fly; (2) path planning could avoid these targets without affecting the flight. A possible problem could occur close to the landing zone, which could be interdicted by a great number of these false obstacles. Nevertheless, this radar system is not intended for operating in landing operations (as the radar is pointed to the front and not pointed down) and could be disactivated during the landing maneuver (when other sensors are operating).

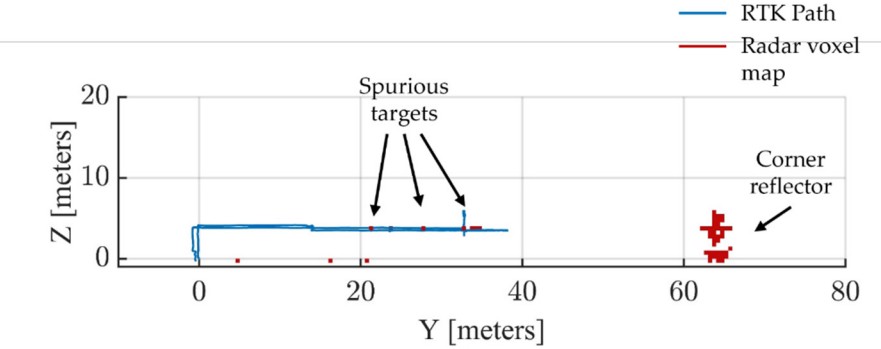

**Figure 14.** Radar map in fixed frame in Y–Z plane.

As mentioned above, the targets are (conventionally) mapped at the same elevation as the flying UAS. This is a rough assumption, but effective. Indeed, because of the poor angular resolution of the radar, the elevation detected by the radar could even worsen the radar mapping, as shown in Figure 15a. Figure 15a was obtained taking into account both the pitch angle (that UAS provides) and vertical positioning of the detected targets provided by the MIMO. Figure 15b was obtained considering only the pitch angle (that UAS provides), but not the vertical positioning of the detected targets provided by the MIMO. In both cases, the result is a sort of a large swipe of the target (a single naval corner reflector).

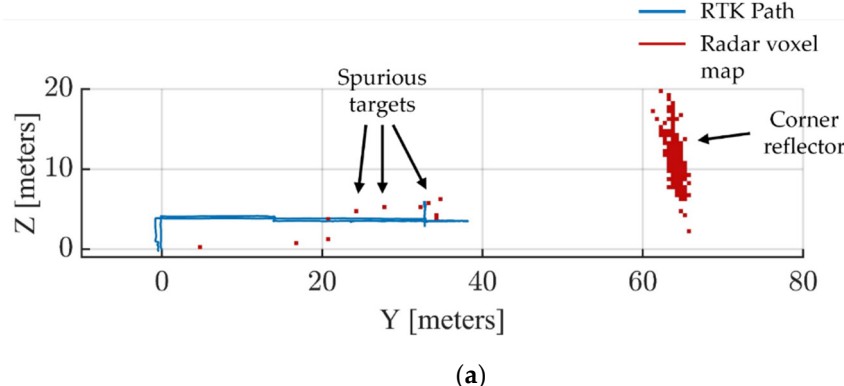

(**a**)

**Figure 15.** *Cont.*

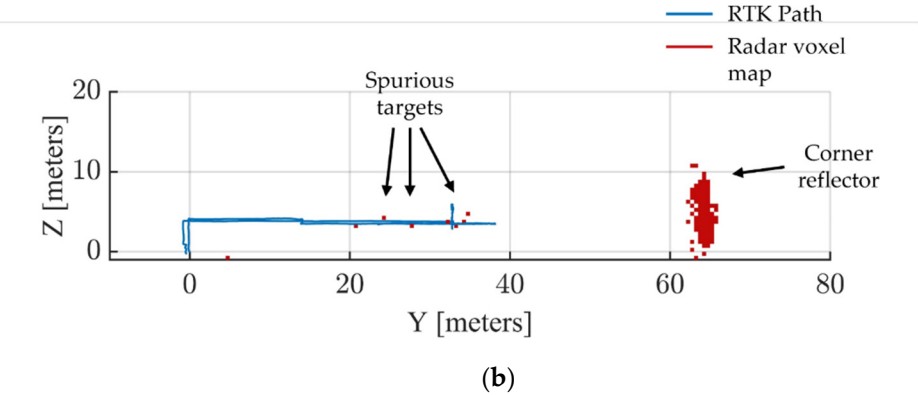

(**b**)

**Figure 15.** Radar map in fixed frame in Y–Z plane obtained: (**a**) using (4) as rotation matrix and considering the elevation detected by the radar, and (**b**) using (4) as rotation matrix and $z_{r,i} = 0$.

*3.2. Wooded Area*

The wooded scenario is shown in Figure 16a. The drone was located about 60 m away from the wood. The path during the test is shown in Figure 16b. We moved towards the wooded area by increasing elevation progressively. When the drone was close to the wood, we moved laterally in order to scan a portion of the wood.

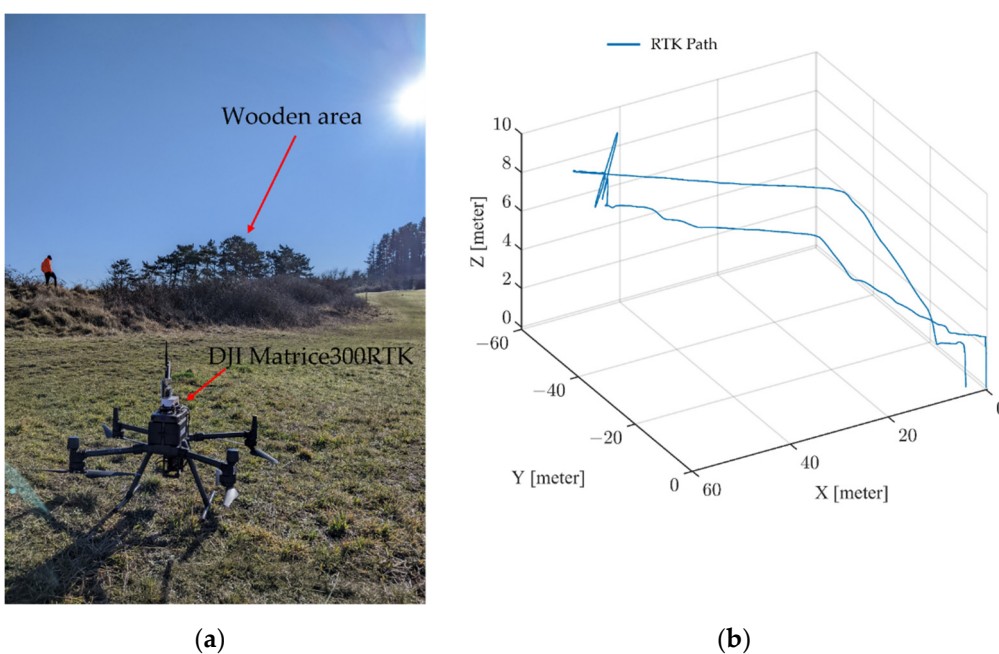

(**a**)                                                                                      (**b**)

**Figure 16.** Picture of the wooded scenario (**a**) and the flight path (**b**).

The mapped area is shown in Figure 17. By comparing the satellite view with the radar map, we can recognize the shape of the wood marked with A. The other sparse targets inside the dotted area correspond to some bushes.

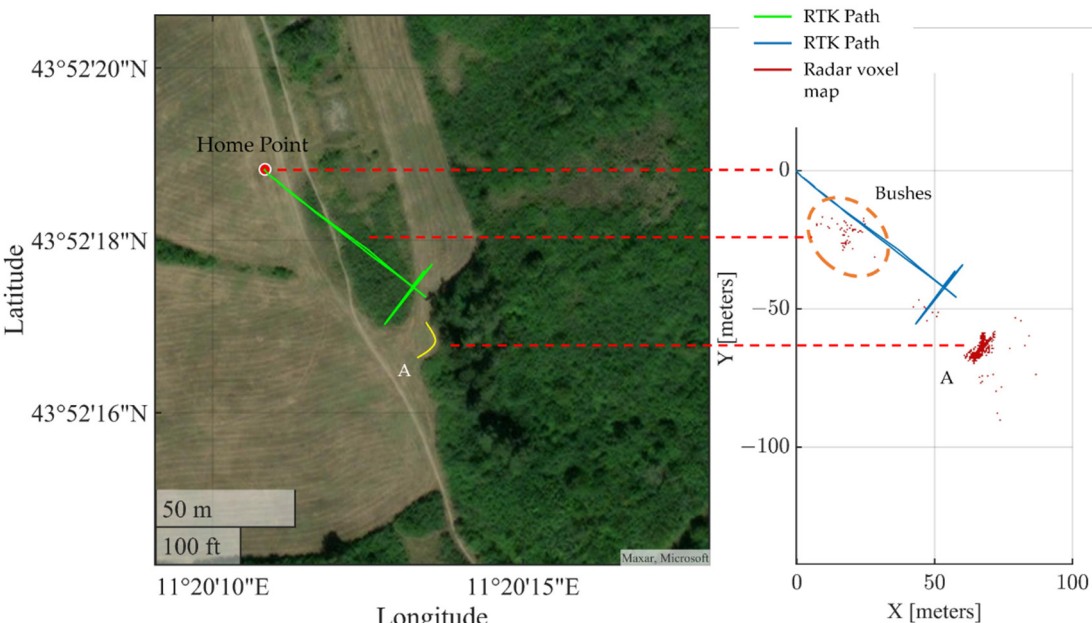

**Figure 17.** Radar map of the wooded scenario seen from above.

The vertical profile of the map is reported in Figure 18. The bushes were located under the flight altitude.

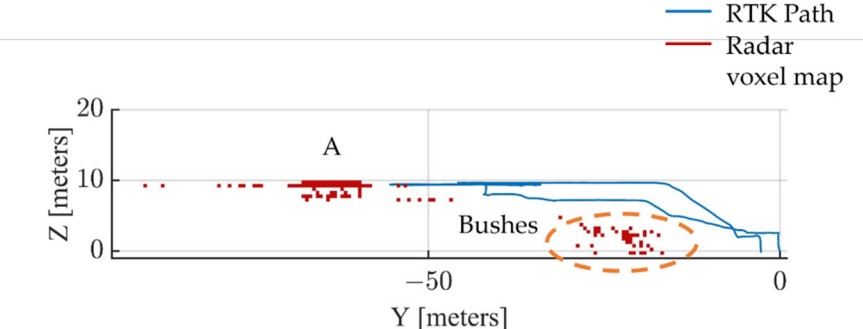

**Figure 18.** Radar map of wooded scenario in Y–Z plane.

### 3.3. Ruined Building

The final test was performed on a ruined building. The ruined complex consists of a main building and two separate courtyard buildings (on the left in Figure 19). We performed two flights in this area. During the first flight the drone flew close to the main building and scanned the front face from left to right. When the drone was at the extremity of its lateral path, we rotated the drone in overing mode. During the second flight, the drone flew closer to the front facade.

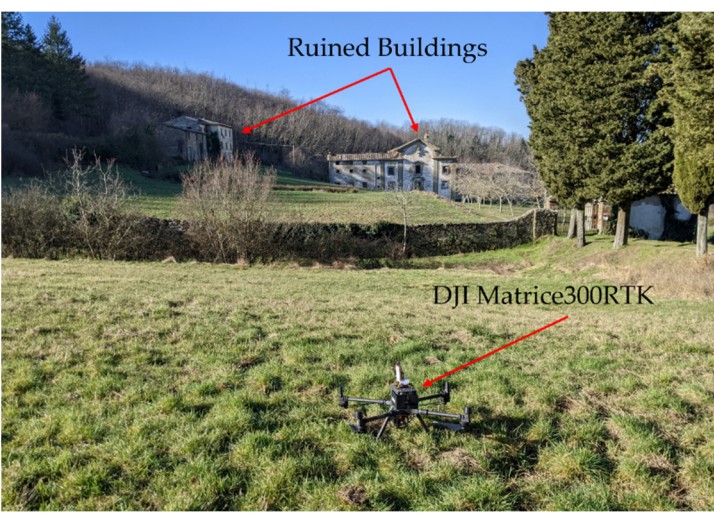

**Figure 19.** Picture of the scenario with a ruined building.

The results of these flights are shown in Figures 20 and 21. The front face of the main building is well clear in both cases. In Figure 20 we mapped a portion of the courtyard buildings.

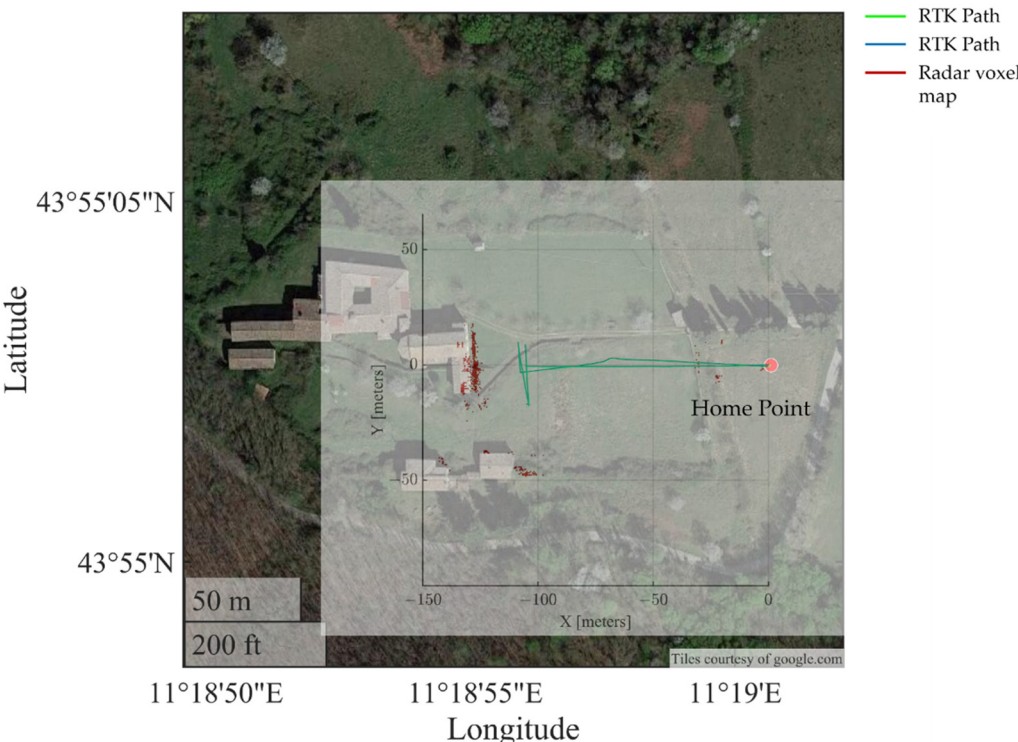

**Figure 20.** Radar map and trajectory superimposed on an aerial picture (from Google Maps).

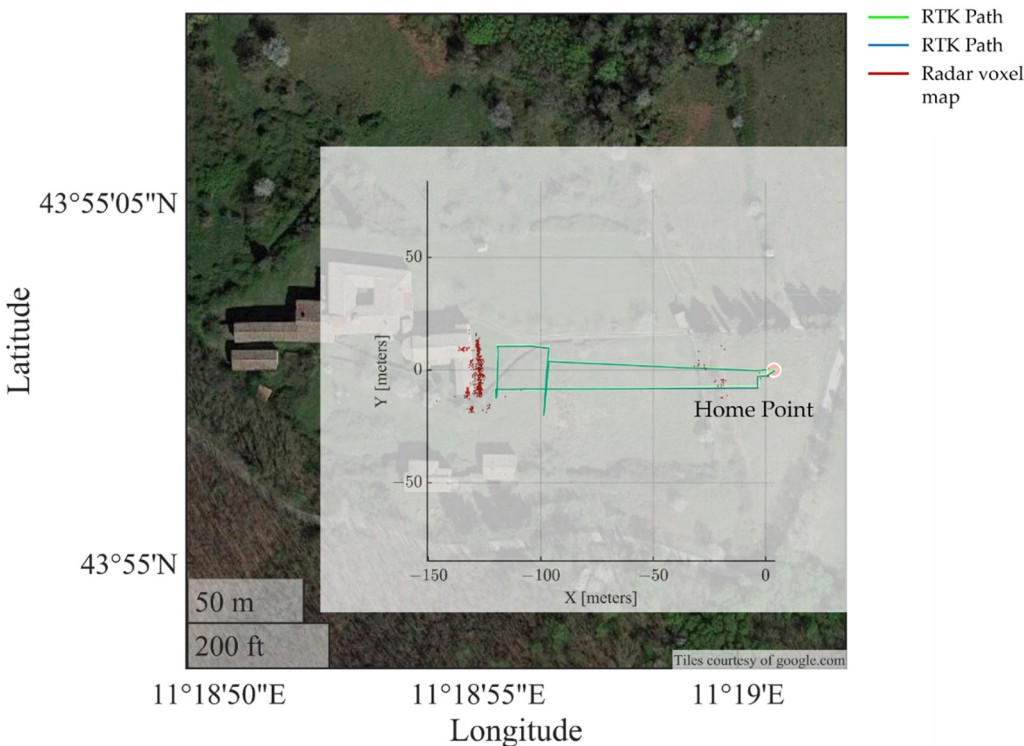

**Figure 21.** Radar map and trajectory superimposed on an aerial picture (from Google Maps).

## 4. Discussion

This is the first attempt to perform entirely radar-based mapping for autonomous UAS flights. Unfortunately, the radar is not able to provide obstacle elevation. This may introduce artifacts, i.e., objects mapped at wrong elevations (Figure 15). To overcome this limitation, the authors propose setting the elevation of detected objects equal to the drone's flight altitude. Thus, the object can be avoided by flying around it or by increasing the drone's altitude (Figure 9).

In scientific literature, the radar was used for terrestrial mapping [10,11], or for UAS obstacle detection and avoidance [12,13], but no one used the same radar for both tasks. This is the main achievement of this work. Other authors [14] proposed data fusion between a monocular camera and radar for obstacle avoidance. The latter is an interesting approach that could be a possible development of the present work. Indeed, the idea we are working on is a system that uses radar for long/medium range detection operations and a visual system based on stereoscopic cameras able to provide a 3D map for short range operations. The two systems should operate in cooperation on the same global map. Figure 22 shows a sketch of the complete system, which integrates data from the radar, vision system, global navigation satellite system (GNSS), and inertial measurement unit (IMU).

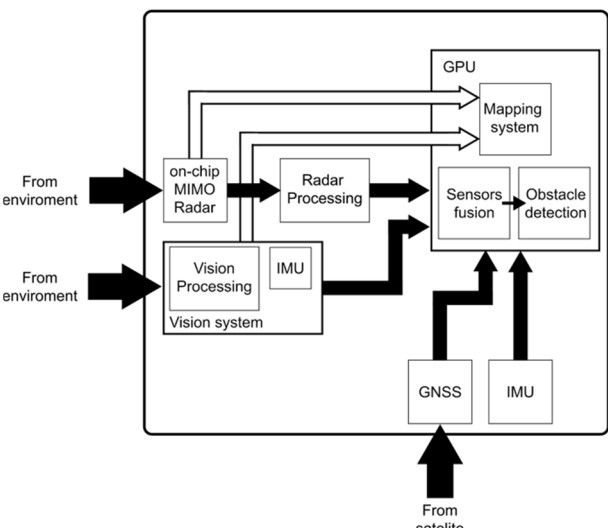

**Figure 22.** Block scheme of the complete obstacle-avoidance and mapping system that integrates radar and visual data.

## 5. Conclusions

In this article the authors presented a mapping system using an anti-collision radar. The proposed method allows the creation of 3D maps of the environment in front of the drone. The coordinates of the objects on the flight plane (left/right position and the frontal distance) were provided by the radar, while the elevation was estimated using the vertical position of the drone.

The mapping system was tested in the following three different scenarios: a single metallic target in a grass, a wooded area, and a ruined building. For each of those scenarios, the map was correctly retrieved. Indeed, the maps are always in good accordance with the satellite view, even if spurious targets appear to be at the same elevation of the flight trajectory, and even if no objects were at that elevation. These false targets are related to ground reflections and disappear by increasing the flight altitude.

By considering the results of this paper, the authors elaborated a possible path-planning method for obstacle avoidance. In particular, they proposed flying sideways or above detected targets. In this second case, the drone must increase flight altitude until the obstacle disappears below the drone. The success of this strategy is particularly visible during the test in the wooded area, when the radar detected some bushes below the drone.

The main achievement of these preliminary tests was to demonstrate the working principle of this radar technique. Nevertheless, effective performances (in terms of range, false alarm probability and undetected targets, etc.) must be assessed in specifically designed experimental tests.

**Author Contributions:** Conceptualization, L.M., L.B., M.P., E.B. and M.B.; methodology, L.B. and L.M.; software, L.B. and T.C.; validation, L.B., L.M. and M.P.; formal analysis, L.M. and L.B.; investigation, L.B., L.M, T.C., M.B., A.B.; resources, L.M., L.B. and T.C.; data curation, L.M.; writing—original draft preparation, L.M. and L.B.; writing—review and editing, E.B., M.P.; visualization, L.M., L.B.; supervision, M.P., M.B.; project administration, M.P.; funding acquisition, M.P. All authors have read and agreed to the published version of the manuscript.

**Funding:** This research was co-funded by Horizon 2020, European Community (EU), AURORA (Safe Urban Air Mobility for European Citizens) project, ID number 101007134.

**Institutional Review Board Statement:** Not applicable.

**Informed Consent Statement:** Not applicable.

**Data Availability Statement:** Data available on request.

**Conflicts of Interest:** The authors declare no conflict of interest.

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
