# Peer review of "Geo-Referenced Mapping through an Anti-Collision Radar Aboard an Unmanned Aerial System"

_drones, doi:10.3390/drones6030072_

Round 1

Reviewer 1 Report

I have reviewed the manuscript entitled "Geo-referenced mapping through an anti-collision radar aboard a UAS".

The manuscript relies on a strong engineering and development background on UAS navigation and proposed a novel radar-based system.

However, the manuscript itself fails seriously as a scientific journal article. The major problem of this manuscript is that it does not contain any discussion chapter to compare the findings of the authors with other scientific literature. This manuscript seems to be more like a technical note that gives a too extensive description of the methodology and then summarizes the main results only. But how can we evaluate these results if there is no valid comparison with any other scientific works on UAS navigation? Unfortunately, in my opinion, this manuscript cannot be published in its recent stage.

My other comment: the Introduction chapter is really short and it seems that the authors have not formulated a clear research gap to be addressed by the manuscript itself. Furthermore, the last paragraph with introducing the funding project and so on is absolutely irrelevant in the body text of a scientific article.

Reviewer 2 Report

Hi,

     This paper summarized an airborne radar that can be installed on a small UAV for sense and avoid applications. The English writing of the paper has to be improved in the revision submission. In general, this paper looks more like a technical report, instead of a journal paper. To be a journal paper, the novelty of the research and the contributions to the research work must be addressed. Methodologies and experiments analyses are also needed to be discussed in the paper clearly. Thus, please have a check a published journal paper, rewrite this paper, and visit the English writing center at the university for proofreading before the revision submission. For detailed comments, please also check the highlighted words listed in the reviewed paper as attached.

Thanks a lot!

Reviewer 3 Report

In this article, the authors propose a radar-based mapping system both for obstacle-detection and for  path-planning. 

Set up
The radar equipment is a single-chip device originally developed for automotive applications that has good resolution in azimuth, but poor resolution in elevation (unfortunately this feature is common to most radars). This is a critical limitation that the UAV has to consider in the selection of obstacle-avoidance maneuvers and in path-planning. 

Results
Nevertheless, the radar mapping system, proposed in this paper, was successfully tested in different scenarios: a single metallic target in a grass, a vegetated scenario, and in proximity of a ruined building.

I agree with the background mentioned by the authors, namely "Unmanned Aerial Systems (UAS), especially in combination with autonomous guidance, have an enormous potential in many fields of application. Nevertheless, an open challenge for safe autonomous flight is to rely on a mapping system for local positioning and obstacle avoidance." The work is relevant and in line with the journal. The work is also clearly written and the demonstrations provided by tha authors are very convincing. At the same time, the work may benefit from referring to other works on MPDI about UAVs, e.g. Multiview image matching of optical satellite and UAV based on a joint description neural network; Coverage path planning methods focusing on energy efficient and cooperative strategies for unmanned aerial vehicles; A semi-physical platform for guidance and formations of fixed-wing unmanned aerial vehicles; Drone technology for monitoring protected areas in remote and fragile environments. Overall, I am ok with supporting with work, another good point is that the figures and plots are very clear and describe the experiments very clearly. I add the following minor comment

- i noticed the authors used a DJI UAV. As far as I know, this type of drone is not perfect for research, since the autopilot cannot be programmed and expanded. This is different than other autopilots that are open source such as ArduPilot, PX4 etc. In other words, how can the authors embed the proposed system in the proprietary DJI architecture?

Round 2

Reviewer 2 Report

Hi,

     The English writing has been improved a lot in this revision. The introduction is pretty good now; however, the main contents of the paper still require further improvements. As commented in the reviewed paper, please discuss your research objectives, achievements and contributions in the paper. The experiment results listed in the paper should be used to support and prove that the research objectives have been achieved, so please modify this paper for the next revision.

Thanks!

Author Response

see pdf attached 
